# Evaluation of Increasing Dietary Concentrations of a Multi-Enzyme Complex in Feedlot Lambs’ Rations

**DOI:** 10.3390/ani14081215

**Published:** 2024-04-18

**Authors:** Germán David Mendoza-Martínez, Pedro Abel Hernández-García, Cesar Díaz-Galván, Pablo Benjamín Razo-Ortiz, Juan José Ojeda-Carrasco, Nalley Sánchez-López, María Eugenia de la Torre-Hernández

**Affiliations:** 1Departamento de Producción Agrícola y Animal, Universidad Autónoma Metropolitana Xochimilco, México City 04960, Mexico; gmendoza@correo.xoc.uam.mx (G.D.M.-M.); cesarwardi14@gmail.com (C.D.-G.);; 2Centro Universitario Amecameca, Universidad Autónoma del Estado de México, Amecameca 56900, Mexico; mvzrazo@gmail.com (P.B.R.-O.);; 3Conahcyt-UAM Xochimilco, Universidad Autónoma Metropolitana Xochimilco, México City 04960, Mexico; mdelatorre@correo.xoc.uam.mx

**Keywords:** exogenous enzymes, feed additives, rumen microbiota, lambs

## Abstract

**Simple Summary:**

Exogenous enzymes can improve domestic ruminants’ productive performance by improving the amount of nutrients obtained from their rations, mainly from the fibrous portions. Dietary inclusion of exogenous enzymes can act synergistically with endogenous enzymes from the rumen during digestion and can modify the populations of ruminal communities and methane emissions. In the present study, dietary supplementation of a multi-enzyme (M-E) complex (Optimax E^®^) at 0, 0.2, 0.4, and 0.8% of dry matter (DM) increased nutrient digestibility, daily weight gain, and net energy, and improved feed conversion and the Firmicutes/Bacteroidetes ratio. Results confirm that including Optimax E^®^ can be used to improve lambs’ performance.

**Abstract:**

The objective of this study was to evaluate the effects of increasing levels of the M-E complex (xylanase, glucanase, cellulase, and invertase) Optimax E^®^ on the performance of growing lambs, their digestibility, and their rumen microbiota, and to estimate NEm, NEg, and ruminal methane levels. Forty lambs (Katahdin x Dorset; 22.91 ± 4.16 kg) were randomly assigned to dietary concentrations of ME (0, 0.2, 0.4, and 0.8% DM) and fed individually for 77 days. Increasing M-E improved feed conversion (*p* < 0.05) as well as NEm and NEg (*p* < 0.05), which were associated with increased in vivo DM and NDF digestion (linear and quadratic *p* < 0.01). Few microbial families showed abundancy changes (Erysipelotrichaceae, Christensenellaceae, Lentisphaerae, and Clostridial Family XIII); however, the dominant phylum Bacteroidetes was linearly reduced, while Firmicutes increased (*p* < 0.01), resulting in a greater Firmicutes-to-Bacteroidetes ratio. Total Entodinium showed a quadratic response (*p* < 0.10), increasing its abundancy as the enzyme dose was augmented. The daily emission intensity of methane (per kg of DMI or AGD) was reduced linearly (*p* < 0.01). In conclusion, adding the M-E complex Optimax E^®^ to growing lambs’ diets improves their productive performance by acting synergistically with the rumen microbiota, modifying the Firmicutes-to-Bacteroidetes ratio toward more efficient fermentation, and shows the potential to reduce the intensity of greenhouse gas emissions from lambs.

## 1. Introduction

The productive response of ruminants to dietary exogenous enzymes depends on several factors such as the type of enzyme, the daily dose, the rumen environment, and the characteristics of forages and diet composition [1], which could explain the variability of responses observed to date [2]. Nevertheless, there are some experiments where increasing doses of exogenous fibrolytic enzymes with low-quality forages showed a linear response in NDF digestion and daily gain in growing steers [3,4], and a similar dose response was observed with fibrolytic enzymes in milk production with dairy cattle [5]. However, there may be a limit where the response to enzyme supplementation will be nonlinear [6,7].

Most of the studies in ruminants have been conducted with cellulases–xylanases [8,9], amylases [8,10,11], or with β-glucanases or hemicellulases [7]. Furthermore, few experiments have evaluated dietary supplementation with phytase [12], protease [13], or glucoamylase [14,15]. It should be noted that there have been studies focusing on the supplementation of enzyme complexes [16,17]. In non-ruminants, M-E preparations are frequently used [18,19], with blends formulated on a ration of indigestible components to obtain more nutrients [20]. Since new enzyme mixtures are being offered to the feed additive market, it is important to identify which enzyme preparations and activities can improve the digestibility of forages of varying quality. For example, corn silage is one of the most widely used forages in ruminants’ diets [21] and, because of its composition (starch and cell walls), could be an excellent candidate ingredient to evaluate the effects of various enzyme preparations.

Although multiple evaluations with exogenous enzymes have reported increases in nutrient digestibility, only a few report changes in indigestible energy [8,22], while others present improvements in metabolizable energy (ME) or net energy (NE) [23,24], information which is important for animal nutritionists to make decisions related to diet formulation. To estimate the dietary energy changes related to enzyme supplementation, the net energy for maintenance (NEm) and gain (NEg) obtained in diets can be estimated from ruminants’ performance, as described with other additives [25,26]. Therefore, the objective of the present experiment was to evaluate increasing doses of a complex in growing lambs fed a ration containing 40% corn silage to estimate its impact on net energy and to evaluate if M-E supplementation could modify ruminal fermentation patterns. The overall hypothesis of this study was that increasing dietary levels of an M-E blend will lead to a linear improvement in daily gain and net energy in lambs, which would be associated with digestibility improvements and a fermentation profile with increased propionate and reduced methane, which in turn would be associated with changes in the bacterial, archaeal, ruminal, fungi, and protozoan families.

## 2. Materials and Methods

The experiment was conducted under guidelines approved by the Academic Committee of the Department of Animal Science of Ethics, Biosafety and Animal Welfare of the UAEM Amecameca University Campus of the Autonomous University of the State of Mexico, at the experimental facilities of the Autonomous University of the State of Mexico in UAEM Amecameca University Campus.

Forty Katahdin × Dorset lambs (initial BW 22.91 ± 4.16 kg) were used in the growth assay, fed in individual pens with four dietary concentrations (0, 0.2, 0.4, and 0.8% DM) of a commercially available M-E, Optimax E^®^ (CBS Bio Platforms, Calgary, AB, Canada), formulated with xylanase (2500 XYL units/g), glucanase (250 GLU units/g), cellulase (1250 CMC units/g), and invertase (1000 Invertase units/g) from *Trichoderma reesei*, *Saccharomyces cerevisiae*, and *Aspergillus aculeatus*. Diets were formulated for growing lambs with NRC [27] (14% CP and 2.56 Mcal ME/kg DM) including 40% corn silage (Table 1). Lambs were dewormed with Closantel (Panamericana Veterinaria de México, Querétaro, Mexico) (5 mg/kg BW), vaccinated against Clostridium perfringens types C and D, *Clostridium novyi, sordeli, chauvoei*, and *septicum* (Ultrabac 7, Zoetis, Mexico City, Mexico), and dosed with vitamins A, D, and E (Vigantol, Bayer, Mexico City, Mexico, 2 mg/lamb) before starting the experiment.

The lambs had continuous access to clean water and were weighed on two consecutive days at the beginning (Days 0 and 1) and the end of the trial (Days 76 and 77). The feed was provided at 0800 h and 1800 h and lambs had free access to the feed, ensuring 100 g orts daily. The daily feed intake, the average daily gain (ADG), and the feed conversion (ratio of kg feed intake/kg gain) were evaluated. Fecal samples were collected for five consecutive days (from Days 63 to 68 of the experimental period) to estimate the apparent DM and NDF digestion using acid-insoluble ash as an internal marker to estimate the DM digestibility [28]. Cell walls were analyzed with the Van Soest procedures [29]. The in vitro indigestible fraction was estimated by incubating rations in vitro at 72 h with ruminal fluid [30].

On Day 77, ruminal fluid was sampled with an esophageal probe in pre-prandial conditions and then the rumen fluid pH was measured using a glass electrode (Orion Star, Model A215, Thermo Scientific™; Waltham, MA, USA) then filtered through four layers of cheesecloth. A subsample of each tube was cryopreserved and stored at −80 °C until DNA extraction. A second subsample was acidified with metaphosphoric acid. The volatile fatty acid (VFA) concentration was measured with gas chromatography (Clarus 580, Perkin Elmer, Waltham, MA, USA), [31] and methane and carbon dioxide were estimated using stoichiometry equations [32].

Cryopreserved ruminal fluid aliquots (250 µL) from 5 lambs per treatment were taken from each sample and metagenomic DNA extraction was performed with the ZymoBiomics^®^ kit (#cat. D4300T; Irvine, CA, USA). The quality and quantity of DNA was evaluated with a NanoDrop 1000 (Thermo Scientific, Waltham, MA, USA). The metagenomic DNA concentrations obtained were among 25.9 ng/µL and 127.7 ng/µL. The DNA samples were sent to ZymoBiomics^®^ (Zymo Research, Irvine, CA, USA) to be sequenced on the Illumina^®^ Nextseq platform (Illumina, Inc, San Diego, CA USA). Microbial community structure analyses of the bacteria and archaea were conducted through analysis of the V3–V4 region of the 16S rRNA gene [33], and changes in the rumen protozoa were analyzed through analysis of the 18S rRNA gene [34].

Targeted sequencing of the 16S and 18S rRNA genes was performed using the Quick-16S™ NGS library preparation kit (Zymo Research, Irvine, CA, USA). The V3–V4 region of the 16S and 18S rRNA genes was amplified with specific primers designed by Zymo Research. The final pooled libraries were cleaned with the Select-a-Size DNA Clean & Concentrator™ (Zymo Research, Irvine, CA, USA), then quantified with TapeStation^®^ (Agilent Technologies, Santa Clara, CA, USA) and Qubit^®^ (Thermo Fisher Scientific, Waltham, WA, USA). As a positive control for each targeted library preparation, the ZymoBIOMICS^®^ microbial community DNA standard (Zymo Research, Irvine, CA, USA) was used. 

The final libraries were sequenced on the Illumina^®^ Nextseq™ platform with a P1 reagent kit (600 cycles). Sequencing was carried out with a 30% PhiX spike-in. Unique amplicon sequence variants were inferred from raw reads using DADA2 *cleavage* [35], which also served to eliminate chimeric sequences. The taxonomy assignment was carried out using Uclust from Qiime v.1.9.1 [36] with Zymo’s own database. Composition and alpha and beta [37] diversity visualization analyses were also performed with Qiime v.1.9.1. Community richness and diversity were analyzed using the Shannon index, the observed species, and Simpson’s reciprocal index [38]. For alpha diversity analysis of the microscopic eukaryotes in the rumen, the observed species as a measure of richness as well as the Shannon and Simpson indices for each treatment were also calculated. 

The observed NEm and NEg in each lamb were estimated from the initial and final body weight (BW), daily dry matter (DMI), and average daily gain, as described by Zinn et al. [25], based on shrunken body weight (SBW) as 96% of the full body weight BW [39]. The maintenance energy requirement (Mcal/d) was estimated from the metabolic BW (ME = 0.056 BW0.75) and gain energy requirement (Mcal/d) with the coefficient 0.276 [40], with the formula energy gain (EG) (Mcal/d) = 0.276 × ADG × SBW0.75. The observed net energy of maintenance (NEm) and net gain energy (NEg) were derived from the maintenance energy (ME), energy gain (EG), and DMI, substituting these values to obtain the constants a = −0.41 × EM; b = 0.877 × ME + 0.41 DMI + EG; and c = −0.877 × DMI [25], which were substituted into the following quadratic formulae: NEm = (−b ± √ (b2 − 4ac))/2c (Mcal/kg) and ENm and NEg = (0.877 ± NEm) − 0.410, as described for lambs by Arteaga-Wences et al. [41]. The effect size of enzymes was also expressed as the percentage relative increase in NEm and NEg compared to the control group. The size effect (SE) estimated net energy was expressed as the percentage change between the enzyme dietary concentration and the controls, and the enzyme average SE was compared to the control with a Chi-squared test using the MedCalc Version 22.003 statistical software (https://www.medcalc.org/manual/chi-square-test.php accessed on 26 February 2024).

The stoichiometric equations proposed by Wolin [42] and simplified by Van Soest [32] were used to estimate moles of CH_4_ and CO_2_ based on the VFAs. Moles of hexose fermented in the rumen were estimated from digestible carbohydrate intake using digestibility values [43]. Methane and carbon dioxide were expressed per unit of intake (grams per kilogram of DMI) and per kg of average daily gain (grams per g/kg of ADG; [44]).

The data of all response variables were tested for normal distribution (Shapiro–Wilk test) and then analyzed as a complete randomized design, testing linear and quadratic effects using coefficients to adjust orthogonality using the Proc IML of SAS (9.04.01, 2022 SAS Inst. Inc., Cary, NC, USA); lamb performance data were analyzed using the initial BW as a covariate. Response variable associations were tested with Pearson correlation coefficients.

## 3. Results

Increasing dietary concentrations of M-E improved ADG (*p* < 0.10) and feed conversion (*p* < 0.05) linearly. There was a linear increment in the NEm and NEg (*p* < 0.05) over the control associated with a linear intake of the enzyme (*p* < 0.001; Table 2), which represented an increment in size depending on the enzyme dietary concentration form from 7.32 to 14.63 in NEm and from 5.35 to 10.7 in NEg (Table 3). The addition of M-E allowed the reduced indigestible fraction with a quadratic effect (*p* = 0.04). In vivo DM and NDF digestibility were increased as the dietary enzyme was augmented (Table 2; linear *p* < 0.01; quadratic *p* < 0.01) and were correlated (r^2^ = 0.77, *p* < 0.0001) with enzyme intake. In addition, net energy values were positively correlated to digestibility (r^2^ = 0.53; *p* = 0.08).

The ruminal pH was not modified by M-E concentration, even when the total VFA concentration was increased quadratically (*p* < 0.10). The molar proportions of propionate (linear *p* < 0.01) and butyrate (quadratic *p* < 0.01) increased and were positively correlated with DM digestibility (r^2^ = 0.56; *p* = 0.009), while carbon dioxide and methane were reduced linearly (*p* < 0.01). The daily emission intensity of methane and carbon dioxide per kg of intake or daily gain was reduced linearly (*p* < 0.01) in response to the M-E dietary concentration (Table 4).

A total of 10,814,916 raw sequences were obtained for the V3–V4 region of the 16S rRNA gene analysis, within a range of 408,992 to 635,314 (average 540,746 sequences per sample), from sequencing 20 samples. After performing quality filtering, as well as chimera detection and elimination (394,947 sequences), the total sequences analyzed were 4,092,385, in a range of 154,979 to 247,526 (average 204,619 per sample). The sequencing depth to describe the ASV-level bacterial and archaeal diversity was evaluated via a rarefaction curve of the observed species of all samples (Appendix A).

Shannon (homogeneity) and Simpson (diversity) indices, as well as the observed species as a richness estimator, were calculated for each treatment (Table 5), having been normalized to 20,000 sequences. Alpha diversity analysis showed no significant difference in the ruminal microbial community as the M-E dietary concentration increased. The ruminal bacterial and archaeal diversity of each group tended to be stable.

A total of 9,975,378 raw sequences were obtained for the 186S rRNA gene analysis, within a range of 419,974 to 563,398 (average 498,769 per sample). After performing quality control analysis, as well as chimera detection and elimination (27,929 sequences), the total sequences analyzed were 4,436,246, in a range of 187,584 to 249,835 (average 204,619 per sample). The sequencing depth to describe the ASV-level eukaryotic diversity was evaluated using a rarefaction curve of the observed species of all samples (Appendix A). The curves of all samples reached a plateau (from both analyses of the 16S and 18S genes), indicating that a sufficient number of sequences had been generated to investigate microbial diversity in the rumen of the studied lambs. 

The microbial communities of the rumen prokaryotes from lambs fed with different M-E dietary concentrations showed similar richness values and homogeneity (Shannon) and diversity (Simpson) indices (Table 5), without differences among treatments. In contrast, rumen eukaryotes from the M-E-fed lambs showed a linear reduction in richness (*p* < 0.05; observed species) and homogeneity (Shannon; *p* < 0.01) indices without differences in diversity (Simpson; Table 5).

Twenty-two bacterial/archaeal families with a relative abundance >0.1% were identified in all samples (Table 6; Appendix A), of which Prevotellaceae, Bacteroidales, Veillonellaceae, and Ruminococcaceae (average 33%, 22%, 11%, and 5%, respectively) were the most abundant. Few microbial families showed abundancy changes due to the M-E (Table 6); however, there was a linear increase in the abundance of Erysipelotrichaceae (*p* < 0.10), Christensenellaceae, and Lentisphaerae (*p* < 0.05), and a quadratic response (*p* < 0.05) in the Clostridial Family XIII abundancy. The dominant phylum, Bacteroidetes, was linearly reduced, and the Firmicutes increased (*p* < 0.01), which resulted in a linear increase in the ratio of Firmicutes to Bacteroidetes (Table 6).

Changes in the ruminal fungi and protozoa’s relative abundance are shown in Table 7. The total Entodinium showed a quadratic response (*p* < 0.10), increasing its abundancy as the enzyme dose was augmented.

## 4. Discussion

Ruminant performance in response to exogenous enzymes depends on several factors. Key among these are interactions between the characteristics of the enzymes, such as their dose, their activity in ruminal conditions, and the ability of the enzymes to resist rumen degradation [1]. Additionally, the substrate plays a key role in the dietary enzyme response according to the basal diet composition, the complexity of the plant cell wall [7], the grain-to-forage ratio [15], and the animal’s genetic potential and its microbiota response [45]. Data from our experiment indicate that the specific M-E evaluated had positive effects in a diet with corn silage as the only forage. The data indicate that cellulases and xylanases within the preparation acted on the cell walls of the corn silage and on the xylan present in the grains [46], and β-glucanase acted on non-starch polysaccharides [47]. This substrate hydrolysis combined to increase the digestible energy, which was confirmed in the estimates of the net energy retained in the lambs. Inconsistent results have been reported with fibrolytic enzymes in finishing sheep [48,49] where the diet has a low proportion of forage and the more acidic conditions of the rumen reduce the digestion of the NDF fraction [1].

Two meta-analyses confirm that fibrolytic enzymes show consistent increases in DM and NDF digestibility with different magnitudes, whereby dry matter digestibility (DMD) increased from 1.3% to 11% for DMD and NDF from 2.30% to 16.55%, respectively [8,50]. The increment observed in NDF digestibility in this experiment is higher than that reported in other experiments with values from 8.0 to 10% [51]; 8.38% [3]; or 11.3% [52]. These NDF digestibility improvements can be explained by the potentially digestible fraction [1] in the corn silage, which has a small indigestible NDF fraction (11.37% [53]), providing energy from starch (25 to 35% starch) and from its NDF (40 to 50% NDF [54]). The increase in digestibility explains the increase in the NEm and NEg values.

Previous experiences with increasing exogenous fibrolytic enzyme doses with tropical forages also showed a linear response in NDF digestion and ADG in growing steers [3,4]. Experiments conducted with sheep with forages of different quality suggest that one of the main factors in the response to exogenous fibrolytic enzymes is the quality of the forage, particularly its potentially digestible fraction [1]. Theoretically, the indigestible fraction (estimated in vitro), should have been similar among treatments; however, the results suggest that exogenous enzymes allow greater access to the cell wall. 

It is important to identify changes in rumen microbial populations to understand the interactions that occur among microorganisms and the effects that occur when supplying exogenous enzymes [55] to achieve the synergistic effect in obtaining the greatest amount of nutrients. Differences between rumen species, enzymes, doses, and forages [55] can cause changes in different microbial families that differ between studies [56]. The Erysipelotrichaceae family appears to be related to lactic acid synthesis in low-methane-emitting sheep [57], and lactic acid synthesis may reduce hydrogen availability for methane production [58]. The abundance of Lentisphaerae was positively associated with weight gain in more efficient steers [59] and was three times more abundant in animals without subacute rumen acidosis [60]. For comparison, the abundance of Clostridial Family XIII was greater in Nellore bulls with low efficiency fed a ration formulated with 61.5% corn silage, while the abundance of Clostridial Family was higher in Charolais cattle with low residual feed intake [61].

In several studies, reference has been made to changes in the ratio between the Firmicutes and Bacteroidetes phyla as it has been associated with the efficiency of nutrient utilization [62,63]. Exogenous enzymes have increased Firmicutes abundance in goats fed tropical forages [64] and in sheep fed buckwheat straw and alfalfa [65]. Even when Bacteroidetes have not been considered to play a dominant role in cellulose degradation [66], the increase in this phylum with enzymes may be due to the release of greater amounts of glucose and xylose that stimulate Bacteroidetes.

Although plant cell walls are degraded by a combination of bacteria, fungi, and protozoa [67], less attention has been paid to rumen eukaryotes communities than to prokaryotic. It is not clear why the fungi showed a variable response to the concentrations of the M-E complex, but it is known that fungi have cellulolytic activity [66] and act synergistically in cell wall digestion by physically disrupting the lignified tissues, allowing the rumen bacteria access to the cell wall structures [67].

Rumen protozoans play an important role in ruminal starch digestion [68] and are also recognized for cellulolytic, hemicellulolytic, and pectinolytic activity [67]. Williams et al. [69], based on results obtained with omics techniques, detected glycosyl hydrolases, polysaccharide lyases, and deacetylases, xylanases, pectinases, mannanases, and chitinases. The greatest information on ciliates is from direct microscopic counts showing that both Entodinomorphs and Holotrichs show great variation in their numbers [70]; nevertheless, Grigorova et al. [71] detected increases in Entodinium in the presence of alternative M-E complexes. The increases observed in the present study in terms of the total Entodiniums could be associated with the fact that, with the added enzymes, there was a greater release of carbohydrates and therefore a greater proliferation of bacteria that, finally, could serve as food for protozoans.

The importance of the relationship between methanogenic and rumen protozoa has been recognized because of the hydrogen transfer among them [72]. However, one meta-analysis revealed that methane emissions were positively associated with total rumen protozoa and isotrichids but not with entodinomorphs [73]. This may partially explain why, without changes in methanogenic archaea abundancy, the M-E reduced the intensity of methane emissions. Similar results have been reported in growing finishing lambs with exogenous fibrolytic enzymes in the intensity expressed per kg DM intake [74].

The changes observed in fermentation patterns, combined with those in digestibility, are important indicators to comprehensively understand the response observed in the lambs. The cellulases, xylanases, and glucanases in the M-E improved nutrient digestibility, whereas the invertase increased the fructose and glucose for rumen microorganisms derived from sucrose. An in vitro experiment showed that sucrose fermentation increases butyrate production, but this effect was not observed in vivo [75,76]; presumably, the presence of invertase could contribute to more monosaccharides forming more propionate. 

The overall results indicate that the multi-enzymatic complex administered acted synergistically with the rumen microbiota, which allowed a greater availability of monosaccharides that led to a more efficient fermentation of the feed, as well as greater digestibility, which in turn affected the synthesis of the biomass without causing a reduction in the rumen pH. This culmination of beneficial outcomes resulted in greater net energy utilization and reduced greenhouse gas emissions in lambs fed the M-E preparation.

## 5. Conclusions

The multi-enzyme complex increased the net energy of the ration linearly, improving the productive performance of the lambs through the associated digestibility improvements. The results indicate that the multi-enzymatic complex acted synergistically with the rumen microbiota, modifying the Firmicutes–Bacteroidetes ratio towards a more efficient fermentation, and show potential for reducing the intensity of greenhouse gas emissions from lambs.

## Figures and Tables

**Table 1 animals-14-01215-t001:** Experimental diets and chemical composition (dry matter basis).

	Multi-Enzyme Dietary Concentration % DM			
	0	0.2	0.4	0.8
Corn silage	40.0	40.0	40.0	40.0
Corn grain	30.0	30.0	30.0	30.0
Sorghum grain	11.0	10.8	11.0	10.7
Soybean meal	16.0	16.0	16.0	16.0
Cane molasses	2.0	2.0	1.6	1.5
Mineral premix ^1^	1.0	1.0	1.0	1.0
Optimax E@	0.0	0.2	0.4	0.8

^1^ Commercial Vitasal Engorda Ovinos^®^ content per kg: Ca 27 g, P 3 g, Mg 0.75 g, Na 6.56 g, Cl 10 g, K 0.05 g, S 42 ppm, Fe 978 ppm, Zn 3000 ppm, Se 20 ppm, Co 15 ppm, vitamin A 35,000 IU, vitamin D 150,000 IU, and vitamin E 150 IU.

**Table 2 animals-14-01215-t002:** Lsmeans of the exogenous enzyme complex dietary concentration on lamb performance and total tract digestibility.

	Multi-Enzyme Dietary Concentration % DM					*p*-Value	
	0	0.2	0.4	0.8	SEM	Linear	Quadratic
Enzyme intake g/d	0.0	2.22	4.39	8.75	0.266	0.0001	0.92
Initial BW kg	23.22	23.56	22.38	22.62	1.532	0.69	0.89
Final BW kg *	42.16	43.50	42.50	44.03	2.117	0.60	0.95
ADG kg *	0.246	0.258	0.262	0.278	0.012	0.07	0.88
Intake DM kg	1.126	1.113	1.097	1.094	0.059	0.68	0.86
Feed conversion	4.70	4.33	4.22	3.99	0.181	0.012	0.42
NEm Mcal/kg	1.23	1.32	1.32	1.41	0.047	0.018	0.73
NEg Mcal/kg	1.87	1.97	1.97	2.07	0.054	0.018	0.71
DM digestibility %	61.75	66.72	68.27	68.56	0.698	0.0001	0.0002
NDF digestibility %	10.90	23.80	31.86	26.98	0.422	0.0001	0.0001
In vitro indigestible DM fraction (72 h) %	37.36	30.50	35.39	34.66	1.021	0.64	0.04

* Model included initial BW as a covariate. SEM: standard error of the mean.

**Table 3 animals-14-01215-t003:** Effect size of the enzyme ME dietary concentration on NEm and NEg over the control in lambs fed a ration 40% corn silage.

	Multi-Enzyme Dietary Concentration % DM			
	0	0.2	0.4	0.8
Effect size over the control				
NEm %	0	7.32	7.32	14.63
NEg %	0	5.35	5.35	10.70

Size effect of enzymes: NEm 9.75% (Chi-squared *p*-value = 0.32), NEg 7.13% (Chi-squared *p*-value = 0.39).

**Table 4 animals-14-01215-t004:** Lsmeans of the exogenous enzyme complex dietary concentration on ruminal fermentation.

	Multi-Enzyme Dietary Concentration % DM				SEM	*p*-Value	
	0	0.2	0.4	0.8		Linear	Quadratic
Ruminal pH	7.12	7.04	7.01	7.07	0.084	0.73	0.36
Total VFA mM	110.73	133.64	121.26	122.01	5.390	0.50	0.07
Acetate %	73.33	72.10	69.96	69.43	2.015	0.15	0.60
Propionate %	24.32	35.12	32.22	33.00	0.885	0.001	0.001
Butyrate %	7.71	9.59	8.44	8.57	0.246	0.39	0.009
CO_2_ % molar	54.31	59.22	55.70	55.82	1.383	0.01	0.15
CH_4_ % molar	34.44	32.06	31.14	30.75	1.000	0.007	0.11
CH_4_ emission intensity (g/kg DM intake)	12.75	11.87	11.53	11.23	0.011	0.0001	0.0001
CO_2_ emission intensity (g/kg gain)	116.87	96.23	94.21	80.52	6.502	0.001	0.32

SEM: standard error of the mean.

**Table 5 animals-14-01215-t005:** Effect of multi-enzyme complex dietary concentration on rumen microbial diversity.

	Multi-Enzyme Dietary Concentration % DM				SEM	*p*-Value	
	0	0.2	0.4	0.8		Linear	Quadratic
	Bacteria and Archaea						
Observed species	715.1	907.3	664.0	719.1	71.64	0.49	0.70
Shannon	7.01	7.35	7.05	7.08	0.220	0.89	0.64
Simpson reciprocal	51.03	51.93	49.96	50.52	9.488	0.94	0.98
	Fungi and Protozoa						
Observed species	54.48	68.12	42.86	47.76	4.643	0.05	0.93
Shannon	2.44	2.64	2.12	1.96	0.165	0.01	0.75
Simpson reciprocal	3.81	4.23	3.37	2.91	0.426	0.70	0.62

SEM: standard error of the mean.

**Table 6 animals-14-01215-t006:** Effects of exogenous enzyme complex on relative abundances of bacterial and archaeal families and Bacteroidetes and Firmicutes phyla in the rumen of lambs.

	Multi-Enzyme Dietary Concentration % DM				SEM	*p*-Value	
	0	0.2	0.4	0.8		Linear	Quadratic
Family							
Methanobacteriaceae	2.82	1.86	2.12	2.78	0.557	0.78	0.20
Bifidobacteriaceae	0.32	0.38	0.44	0.36	0.165	0.80	0.81
Coriobacteriaceae	0.32	0.38	0.44	0.36	0.132	0.85	0.55
Bacteroidales	19.92	25.04	21.88	22.52	4.155	0.84	0.65
Prevotellaceae	38.5	36.44	27.4	30.04	4.443	0.14	0.33
Rikenellaceae	1.6	2.84	1.08	0.94	0.665	0.21	0.62
Anaerolineaceae	0.14	0.38	0.4	0.42	0.174	0.33	0.46
Gastranaerophilales	0.08	0.2	0.02	0.0	0.058	0.12	0.59
Fibrobacteraceae	1.24	1.22	1.56	0.92	0.410	0.61	0.43
Christensenellaceae	0.58	1.14	2.18	2.48	0.569	0.02	0.44
Clostridial Family XIII	0.26	0.56	0.42	0.30	0.071	0.66	0.02
Lachnospiraceae	7.62	10.1	10.82	9.26	1.539	0.57	0.17
Clostridial	0.14	0.12	0.26	0.14	0.065	0.81	0.29
Ruminococcaceae	5.66	4.3	5.84	5.44	0.912	0.83	0.80
Erysipelotrichaceae	1.78	1.62	3.1	3.14	0.645	0.08	0.70
Acidaminococcaceae	2.24	1.36	1.42	1.56	0.378	0.35	0.17
Veillonellaceae	10.94	6.64	14.22	13.34	2.326	0.18	0.91
Lentisphaerae	0.36	0.58	0.54	1.38	0.332	0.04	0.50
Desulfovibrionaceae	0.2	0.1	0.06	0.08	0.052	0.15	0.17
Succinivibrionaceae	1.1	1.38	0.28	0.92	0.430	0.52	0.40
Spirochaetaceae	1.36	1.74	1.52	0.86	0.678	0.50	0.53
Mycopkasmataceae	0.12	0.02	0.28	0.2	0.0728	0.18	0.60
Phyla							
Actinobacteria	0.62	0.46	0.84	0.48	0.2565	0.94	0.64
Bacteroidetes	60.14	64.52	50.52	53.66	2.636	0.02	0.17
Fibrobacteres	1.24	1.22	1.56	0.92	0.410	0.61	0.43
Firmicutes	29.22	25.84	38.26	35.66	2.316	0.01	0.37
Lentisphaerae	0.36	0.58	0.54	1.38	0.332	0.04	0.50
Bacteroidetes	60.02	64.32	50.36	53.50	2.636	0.01	0.17
Proteobacteria	1.46	1.56	0.52	1.08	0.4213	0.41	0.31
Spirochaetae	1.36	1.74	1.52	0.86	0.6788	0.50	0.53
Tenericutes	0.22	0.30	0.40	0.26	0.0728	0.18	0.60
Euryarchaeota	2.84	1.88	2.14	2.78	0.5574	0.78	0.20
Ratio of Firmicutes to Bacteroidetes	0.50	0.40	0.79	0.66	0.069	0.008	0.29

SEM: standard error of the mean.

**Table 7 animals-14-01215-t007:** Effects of rumen exogenous enzyme complex on the relative abundances of ruminal fungi and protozoa.

	Multi-Enzyme Dietary Concentration % DM				SEM	*p*-Value	
	0	0.2	0.4	0.8		Linear	Quadratic
Ruminal fungi							
Orpinomyces	0.04	0.5	0.02	0.12	0.101	0.58	0.32
Ruminal protozoa							
Subphyla Ciliophora							
Parabasalia	0.0	0.18	0.0	0.06	0.050	0.99	0.50
Subclass Trichostomatia							
Haptoria	3.86	3.6	4.64	2.08	1.801	0.50	0.53
Entodinium Canadian Arcott	12.24	18	25.28	20.4	6.712	0.39	0.31
Entodinium	43.43	59.52	45.92	60.84	7.680	0.23	0.97
Other Trichostomatia	7.20	10.18	21.28	12.9	5.867	0.42	0.20
Unidentified protozoa	4.32	7.9	2.8	3.54	1.196	0.19	0.68
Total Entodinium	55.67	77.52	71.20	81.24	8.185	0.26	0.08

SEM: standard error of the mean.

## Data Availability

The datasets used and analyzed during the current study are available from the corresponding author on reasonable request. The data are not publicly available due to restrictions on institutional privacy. The datasets generated for this study can be found at https://www.ncbi.nlm.nih.gov/sra/PRJNA1086969 (accessed on 14 March 2024).

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
