# Peer review of "Evaluation of Increasing Dietary Concentrations of a Multi-Enzyme Complex in Feedlot Lambs’ Rations"

_animals, 2024, doi:10.3390/ani14081215_

Round 1
Reviewer 1 Report
Comments and Suggestions for Authors
row 29 - you remove one parenthesis in ((linear and quadratic…)
row 34 - you include “the multi-enzyme complex” before …Optimax E® or include Optimax E® after “the multi-enzyme complex” in raw 24.
row 63 - For the first time, the extended version should be written and the acronym in brackets, then: Net Energy for maintenance (ENm) and gain (EMg)
row 79 - I think that the genetic type was Katahdin X Dorset and not “Katahdin and Dorset”.
Row 81 - you include one parenthesis in (CBS Bio ……
Row 157 - I don’t understand the bibliographical reference. In the text is written the name of a software while the reference 31 is an article.
Row 167 – you change the https address with the version and year of reference of the SAS software.
Row 173 - I don’t understand: “the average size effect of 9.67% in NEm and 7.29 in NEg”, as written in the text, or “9.75% in NEm and 7.13 in NEg” as do you write in under line of table 3?
Row 174 I think that you write about “potentially indigestible fraction”. You check it.
Row 176-177 – 194 - you write about the correlation, but the correlation table is not shown in the text. You explain why you decided not to report the correlation table in the materials and methods section or, you add the correlation table.
Row 193 – add quadratic in (P<0.10)
Row 209 – Probability you write about table 5 no table 4. Change table number.
Row 210 and 225 and 227 – delete (P>0.05)
Row 248 - in the materials and methods section nothing is written about cubic ratios, so I would remove the sentence about ruminal fungi.
Row 270 - the extended version should be written and the acronym in brackets: DMD
Row 271 and 272 - I think the observed increase in NDF digestibility in this experiment is greater than in other experiments (starting at 10.90 and ending at 26.98 (16.08 percentage points).
Row 309 – delete point
You must add the correlation table as reported in the materials and methods section
Author Response
Dear Reviewer.
We appreciate your comments and suggestions about the document.
In the attached file you will find the responses to the comments you made to us.

Reviewer 2 Report
Comments and Suggestions for Authors
Dear Editor
The manuscript "Evaluation of increasing dietary concentrations of an enzyme complex in feedlot lambs’ rations" is well planned research that described the efficacy of an enzyme complex on improved digestibility of DM & NDF.
General Comments:
Is there any specific reason to use the product "Optimax E®" as multi enzyme complex? Why the Authors have not formulated their own enzyme complex using individual enzymes? Or the Authors are suggested to use words as "commercially available multi-enzyme product".
Introduction and Methods section is very well described.
Results: It is suggested to add the supplementary figure in the main file as it shows the diversity (as described in Table 6, 7).
Discussion section is too long, it could be shortened.
The number of references could be reduced to 40-50 by excluding un-necessary or old references.
Thanks and Regards
Author Response
Dear Reviewer
The authors appreciate the comments which are reflected in the document, and the responses to the comments are also included in the attached file.

Reviewer 3 Report
Comments and Suggestions for Authors
The presented manuscript is very complete and adds valuable information to the animal nutrition field. Both introduction and materials and methods are very complete and detailed. The results are clearly stated and discussion is in accordance to the results. I have no comments other than to congratulate the authors on this manuscript and propose that it is accepted for publication.
Author Response
Dear Reviewer.
The authors of the document sincerely appreciate your comments.